# Enhancing epidemic forecasting with a physics-informed spatial identity neural network

**Satoki Fujita** [ID]*, **Tatsuya Akutsu**

Bioinformatics Center, Institute for Chemical Research, Kyoto University, Uji, Kyoto, Japan

* fujita.satoki.86j@st.kyoto-u.ac.jp, satoki.fujita@gmail.com

## Abstract

Forecasting the future number of confirmed cases in each region is a critical challenge in controlling the spread of infectious diseases. Accurate predictions enable the proactive development of optimal containment strategies. Recently, deep learning-based models have increasingly leveraged graph structures to capture the spatial dynamics of epidemic spread. While intuitive, this approach often increases model complexity, and the resulting performance gains may not justify the added burden. In some cases, it may even lead to overfitting. Moreover, infectious disease data is typically noisy, making it difficult to extract infectious disease-specific dynamics from data without guidance based on epidemiological domain knowledge. To address these issues, we propose a simple yet effective hybrid model for multi-region epidemic forecasting, termed Physics-Informed Spatial IDentity neural network (PISID). This model integrates a spatio-temporal identity (STID)-based neural network module, which encodes spatio-temporal information without relying on graph structures, with an SIR module grounded in classical epidemiological dynamics. Regional characteristics are incorporated via a spatial embedding matrix, and epidemiological parameters are inferred through a fully connected neural network. These parameters are then used to govern the dynamics of the SIR model for forecasting purposes. Experiments on real-world datasets demonstrate that the proposed PISID model achieves stable and superior predictive performance compared to baseline models, with approximately 27K parameters and an average training time of 0.45 seconds per epoch. Additionally, ablation studies validate the effectiveness of the neural network's encoding architecture, and analysis of the decoded epidemiological parameters highlights the model's interpretability. Overall, PISID contributes to reliable epidemic forecasting by integrating data-driven learning with epidemiological domain knowledge.

## Introduction

Infectious diseases have long been intertwined with daily human life, with outbreaks historically causing significant disruptions to public health, society, and the economy.

**Data availability statement:** The data that were used in this study are publicly available from the Ministry of Health, Labour and Welfare (https://www.mhlw.go.jp/stf/covid-19/open-data.html), the Japan LIVE Dashboard (https://github.com/swsoyee/2019-ncov-japan), and the Johns Hopkins Coronavirus Resource Center (https://github.com/CSSEGISandData/COVID-19).

**Funding:** The funder provided support in the form of salaries for author SF, but did not have any additional role in the study design, data collection and analysis, decision to publish, or preparation of the manuscript. The work of TA was supported in part by Japan Society for the Promotion of Science (JSPS), Japan, under Grant 22H00532 and Grant 22K19830. JSPS had no role in study design, data collection and analysis, decision to publish, or preparation of the manuscript. JSPS website: https://www.jsps.go.jp/ The specific roles of the authors are articulated in the 'author contributions' section.

**Competing interests:** SF is an employee at Shionogi & Co. There was no involvement of Shionogi & Co. in the publication process. This does not alter our adherence to PLOS ONE policies on sharing data and materials. The other author declares no conflicts of interest.

For instance, the novel coronavirus disease (COVID-19) has triggered a global pandemic since 2019, resulting in widespread infections and fatalities, and severely impairing social functions [1]. Addressing the threat of such diseases requires accurate epidemic forecasting to enable policymakers to implement timely preventive measures and allocate medical resources effectively.

Many mathematical models for epidemic forecasting have been studied and proposed so far. In recent years, deep learning-based approaches have gained attention due to their strong representational power and predictive accuracy. In particular, because infectious diseases like COVID-19 spread across regions primarily through human mobility, spatio-temporal models incorporating graph neural networks (GNNs) have been developed to capture the spatial dynamics of epidemics. These models extract useful features by modeling dynamic interactions between regions over time, thereby enhancing prediction accuracy. However, learning graph structures, which is a common component of these models, is inherently challenging [2] and increases the model complexity. The increased complexity often leads to reduced computational efficiency and, in some cases, even diminished predictive performance. Moreover, some models rely on auxiliary data such as population mobility [3] or social connectivity [4], to learn graph structures. However, such data are often difficult to obtain and may introduce unintended biases. In addition to the challenges of learning graph structures, the inherent complexity of epidemic dynamics—characterized by exponential transmission dynamics and influenced by diverse factors such as public awareness, climate, and drug availability—exposes deep learning models to the risk of overfitting in exchange for their flexibility in adapting to historical data. On the other hand, classical compartmental models such as the SIR model [5] and its variants, which describe epidemic processes using differential equations, are often employed due to their simplicity and interpretability. These models typically adjust their parameters to best fit historical data. However, this approach cannot adequately account for the inherent uncertainties in future epidemic trends.

Recently, several studies [3,6–8] have attempted to incorporate epidemiological domain knowledge—specifically, physics-informed compartmental models unique to infectious diseases, such as the SIR model—into deep learning frameworks to enhance forecasting accuracy. By incorporating deterministic epidemic dynamics into model architectures or loss functions, these approaches guide neural networks in accordance with the underlying principles of disease transmission—efforts to embed physical laws into neural networks have gained attention, including in disciplines such as the natural sciences [9]. However, they often require the number of individuals in the infectious state at each time point as input, which is typically estimated from the number of newly recovered cases. Such data are generally more difficult to track than the number of newly confirmed cases and are often unavailable. To address scenarios where such detailed data are lacking, we propose a simple and practical physics-informed deep learning model for forecasting the future number of confirmed cases, relying solely on historical confirmed case data and population data. Our model, named the Physics-Informed Spatial IDentity neural network (PISID), integrates the SIR model into a deep learning framework based on STID [10], a spatio-temporal

identity model that avoids the complexity of graph structure learning. Epidemiological parameters are estimated using simple Multi-Layer Perceptron (MLP) layers, incorporating spatial characteristics through a spatial embedding matrix. Based on these parameters and the number of confirmed cases, the number of infectious individuals required for applying the SIR model is inferred. The future number of confirmed cases is then predicted using update equations derived from the infection dynamics in the SIR model. This approach enables interpretable forecasting grounded in epidemiological principles—an aspect often lacking in conventional deep learning models. In summary, the contributions of our study include the following:

- We propose a novel multi-region epidemic forecasting model that leverages epidemiological domain knowledge by combining a classical dynamical system in epidemiology with simple neural networks incorporating region-specific embeddings, without relying on graph structure learning.

- By estimating and utilizing epidemiological parameters, our model enhances interpretability and can describe epidemiological dynamics without requiring additional data on the number of infectious individuals.

- We conduct extensive experiments using real-world COVID-19 data, demonstrating the model's stable predictive performance and interpretability.

The remainder of this paper is organized as follows: the "Related Works" section reviews related works, the "Methodology" section details the proposed model structure, the "Experimental Study" section presents the experimental results, and finally, the "Conclusion" section summarizes the work and discusses directions for future research.

## Related works

Numerous mathematical models have been developed for infectious disease epidemic forecasting, which can broadly be categorized into two groups: traditional mathematical models and deep learning models. Among traditional models, classical compartmental models and their variants are particularly prevalent. In these models, the population is divided into homogeneous subgroups representing different states, and the transitions between these states are typically described by differential equations. The SIR model [5], which classifies individuals as susceptible, infectious, or recovered, is the most fundamental. Variants such as the SEIR model [11], which includes an exposed state, and the SIS model [12], which assumes reinfection in possible, have also been widely studied. These models are often used to gain insights into disease characteristics and to explore future prevention strategies through simulation and parameter estimation. Batistela et al. [13] proposed a compartmental model that accounts for temporary immunity due to infection or vaccination, as well as unreported infections, and evaluated the effects of vaccination and social isolation. Fudolig and Howard [14] developed an SIR model incorporating multiple virus strains to explore the conditions under which endemic equilibrium can occur. Typically, future epidemic dynamics are simulated using parameters either optimized from historical data or manually set. However, this approach cannot account for changes in epidemic characteristics during the forecast period, raising concerns about cumulative errors over multiple time steps. Beyond compartmental models, other traditional mathematical models have also been employed for epidemic forecasting. Achrekar et al. [15] used an Autoregressive Moving Average (ARMA) model to predict future influenza-like illness (ILI) cases based on Twitter message trends. Wang et al. [16] developed a generalized Vector Autoregressive (VAR) model to forecast COVID-19 cases in the United States. The spread of infectious diseases exhibits non-stationary characteristics, influenced by various factors such as changes in viral properties, shifts in human behavior, and advancements in medical care. Therefore, the data distribution may evolve over time. In traditional models such as those mentioned above, which assume strong stationarity, it is particularly important to detect the points at which the distributional properties change. While known events such as lockdowns can be used to define these change points, there have also been efforts to identify unknown change points using a Bayesian approach [17], a genetic algorithm [18], and other techniques [19,20]. In other fields of natural science, a method for handling non-stationarity has been proposed using Bayesian compressive sensing [21].

To address the complex nonlinear relationships that traditional models struggle with, more flexible machine learning models have also been explored for prediction. Battineni et al. [22] conducted COVID-19 outbreak forecasting based on Fb-Prophet, a time series prediction model developed by Facebook that accounts for seasonality and holidays. Sadig et al. [23], on the other hand, employed LightGBM and XGBoost—representative gradient boosting algorithms based on decision trees—to predict the number of COVID-19 cases in real-time scenarios. Deep learning-based approaches that adaptively learn feature representations have also attracted significant attention. ArunKumar et al. [24] applied a Recurrent Neural Network (RNN) to forecast COVID-19 cases, while Lee et al. [25] used a Convolutional Neural Network (CNN) for ILI prediction. Transformer-based models such as Autoformer [26] have also been employed to capture temporal dependencies in time series data. While these models effectively process sequential data, it is important to note that infectious diseases inherently spread through spatial interactions. Consequently, graph-based deep learning methods have attracted attention for modeling spatial dependencies between regions. By representing each region as a node and connecting related regions with edges, Graph Neural Networks (GNNs), such as Graph Convolutional Networks (GCNs) [27], can efficiently capture spatial relationships. Basic graph construction methods often rely on prior knowledge, such as geographic distance or adjacency. For example, Panagopoulos et al. [28] constructed a graph based on human mobility data and analyzed the correlation between population movement and COVID-19 spread across countries. However, such predefined graph structures may not accurately reflect true dependencies. To address this, graph representation learning methods that adaptively learn graph structures using trainable node embeddings have been proposed [29]. ColaGNN [30] extracts inter-regional correlations from temporal latent representations using attention mechanisms, while EpiGNN [31] adaptively learns non-bidirectional spatial correlations that consider both geographical and temporal dependencies, as well as local and global transmission risks. Dual-Topo-STGCN [8] incorporates correlations between geographically distant regions by introducing functional topology that accounts for socio-economic interrelationships, in addition to geographical topology. M-Graphormer [32] learns dynamic graph representations primarily from human mobility data employing three encoding strategies that focus on centrality, spatial characteristics, and edge features. Recently, spatio-temporal GNN models equipped with graph representation learning and incorporating epidemiological domain models such as the SIR model have been proposed [3,6–8]. These models enhance predictive performance by grounding predictions in the physical laws governing disease transmission. However, they require as input the number of infectious individuals at each time point to accurately model infection dynamics. In other words, it is necessary to track not only the number of newly infected individuals who have become infectious, but also those who have ceased to be infectious (i.e., recovered cases) at each time point. Compared to data on new infections, data on recovered cases are often more difficult to follow up on and may be unavailable, which limits the applicability of these models. From a practical standpoint, it is therefore necessary to develop an epidemiologically informed model that can operate solely based on the number of new infections. Furthermore, the inherent complexity of the graph representation learning adopted by the aforementioned spatio-temporal GNN models may hinder performance improvements commensurate with the complexity of the neural architectures themselves [2]. As alternatives that do not rely on graph representation learning, STNorm [33] distinguishes dynamics by normalizing raw data separately in the temporal and spatial dimensions through factorization, while STID [10] ensures spatio-temporal identifiability by embedding temporal identities shared across similar cycles and spatial identities shared within the same region. Despite not utilizing graph representation learning, these models achieve predictive performance comparable to or better than more complex spatio-temporal GNN models. Motivated by these studies, we propose a simple yet effective neural network model that integrates the SIR model into STID, enabling it to capture spatial distinctions without relying on graph representation while also leveraging the underlying epidemiological dynamics. Furthermore, by incorporating a mechanism to infer the number of infectious individuals at each time point based on a simple equation rewrite, our model overcomes the limitation of previous epidemiology-based neural models that required this information as auxiliary input.

## Methodology

In this section, we define the problem setting addressed in this study and present the framework of the proposed model.

### Problem setting

In this study, we address the problem of forecasting the numbers of new confirmed cases in multiple regions based on historical data. Let $X_T = [x_{1,T}, x_{2,T}, \ldots, x_{M,T}] \in \mathrm{R}^M$ denote the number of new confirmed cases in $M$ regions at time step $T$, and let $X_{T-Tin+1:T} = [X_{T-Tin+1}, X_{T-Tin+2}, \ldots, X_T] \in \mathrm{R}^{M \times Tin}$ represent the historical data from time step $T$ going back $T_{in}$ steps. The objective is to forecast the number of new confirmed cases $T_{out}$ steps into the future, denoted $Y_{T+Tout} \in \mathrm{R}^M$, which can be formulated using a mapping function $F$ as follows:

$$Y_{T+T_{out}} = \mathcal{F}(X_{T-T_{in}+1:T}) \tag{1}$$

### Model structure

The overall structure of the proposed model is illustrated in Fig 1 and consists of two modules: a spatio-temporal neural network module and an SIR module. The spatio-temporal neural network module encodes temporal and spatial information based on the historical data of each region and predicts parameters that characterize the underlying epidemiological dynamics. Subsequently, the SIR module forecasts the future number of new confirmed cases by iteratively executing a discrete SIR model using the predicted parameters, leveraging epidemiological domain knowledge.

### Spatio-temporal neural network module

We design a neural network to estimate epidemiological feature parameters. To avoid the potential introduction of erroneous biases caused by overly complex graph representation learning, we construct our framework based on STID [10], a simple yet effective spatio-temporal model. First, the historical input data $X_{T-Tin+1:T} \in \mathbb{R}^{M \times Tin}$ is embedded into a latent space $H_T \in \mathbb{R}^{M \times D}$ from a temporal perspective using a fully connected layer FC($\cdot$) as follows:

$$H_T = \mathrm{FC}(X_{T-T_{in}+1:T}) \tag{2}$$

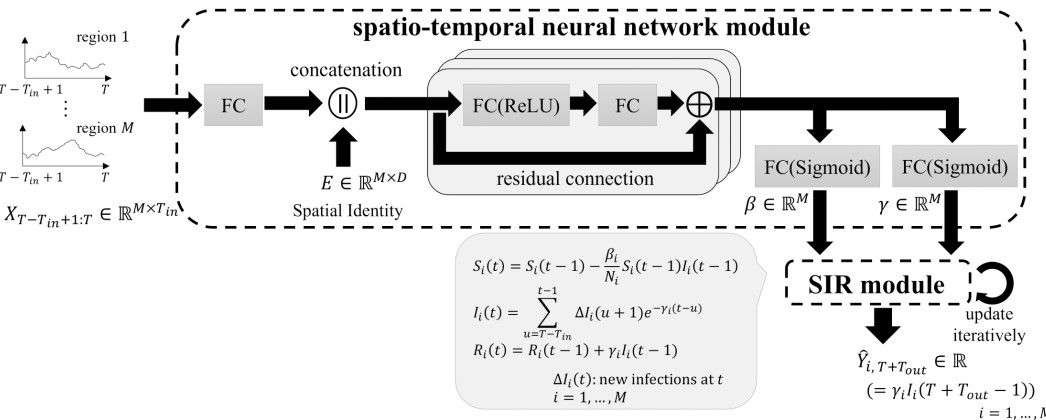

**Fig 1. The entire framework of PISID.**

where $D$ represents the hidden dimension. Next, spatial information is embedded using spatial identities $E \in \mathbb{R}^{M \times D}$, a randomly initialized learnable matrix that captures region-specific features without relying on graph representation learning. The concatenated embeddings $Z^1_T \in \mathbb{R}^{M \times 2D}$, which incorporate both spatial and temporal information, are then used as input to the encoder:

$$Z^1_T = H_T \parallel E \tag{3}$$

The encoder consists of $L$ layers of basic MLP with residual connections:

$$Z^{l+1}_T = FC^l_2 \left( ReLU \left( FC^l_1 \left( Z^l_T \right) \right) \right) + Z^l_T \tag{4}$$

where $FC^l_1$ and $FC^l_2$ with $l \in [1, L]$, denote the first and second fully connected layers of the $l$-th layer, respectively, and ReLU represents the Rectified Linear Unit (ReLU) activation function, applied with dropout. Then, the epidemiological parameters $\beta = [\beta_1, \beta_2, \ldots, \beta_M] \in R^M$ and $\gamma = [\gamma_1, \gamma_2, \ldots, \gamma_M] \in R^M$ are output through FC layers and passed to the SIR module.

$$\beta = Sigmoid \left( FC_\beta \left( Z^{L+1}_T \right) \right), \quad \gamma = Sigmoid(FC_\gamma \left( Z^{L+1}_T \right)) \tag{5}$$

where $FC_\beta$ and $FC_\gamma$ denote the fully connected layers used to estimate $\beta$ and $\gamma$, respectively, and Sigmoid refers to the sigmoid activation function.

## SIR module

The SIR module outputs the target forecast values of the number of new confirmed cases in the future, based on the dynamics of the SIR model. The SIR model is described by the following system of differential equations:

$$\frac{dS_i(t)}{dt} = -\frac{\beta_i}{N_i} S_i(t) I_i(t),$$

$$\frac{dI_i(t)}{dt} = \frac{\beta_i}{N_i} S_i(t) I_i(t) - \gamma_i I_i(t),$$

$$\frac{dR_i(t)}{dt} = \gamma_i I_i(t), \tag{6}$$

where $S_i$, $I_i$, and $R_i$ represent the number of susceptible, infectious, and recovered individuals in region $i$, respectively, and $N_i = S_i(t) + I_i(t) + R_i(t)$ denotes the total population in region $i$. In Equation (6), the infection rate $\beta_i$ and the recovery rate $\gamma_i$ are key parameters that govern the dynamics of disease transmission. The index $R_e(t) := \beta_i / \gamma_i \cdot S_i(t) / N_i$ can be interpreted as the effective reproduction number, which represents the expected number of new infections caused by a single infectious individual in a partially immune population at time step $t$. This metric is often used as a timely indicator of the extent of disease transmission.

We now explain how the aforementioned SIR model is adapted for the current task, which involves forecasting new confirmed cases. These cases are typically assumed to be isolated or treated at the time of reporting and are therefore no longer capable of transmitting the infection. Accordingly, based on the discretized version of Equation (6), the number of new confirmed cases $x_{i,t}$ in region $i$ at time step $t$ can be interpreted as the number of new transitions into the recovered state:

$$\gamma_i I_i(t-1) = x_{i,t} \tag{7}$$

Therefore, our goal is to estimate $\gamma_i I_i(T+T_{out}-1)$. To achieve this, we iteratively update the number of individuals in each compartment over the time interval from $T$ to $T+T_{out}$ based on the dynamics defined by the discretized SIR model. As a first step, we need to determine the initial values of these iterations in each compartment using the available historical data $x_{i,T-Tin+1}, \ldots, x_{i,T}$. The number of recovered individuals $R_i(t)$ can be computed by accumulating the number of new confirmed cases up to time step $t$. Regarding the number of susceptible individuals $S_i(t)$, there is the relation $S_i(t) = N_i - I_i(t) - R_i(t)$. Therefore, once $I_i(t)$ is estimated, the number of individuals in each compartment can be determined, allowing the initial values for the iterations to be set. The differential equation for $I_i(t)$ in Equation (6) can be reformulated as follows [34]:

$$I_i(t) = I_i(0)e^{-\gamma_i t} + \int_0^t \frac{\beta_i}{N_i} S_i(u) I_i(u) e^{-\gamma_i(t-u)} du \tag{8}$$

By approximating the integral in the second term with a discrete summation and treating unavailable data points prior to time step $T - T_{in}$ as negligible, we estimate $I_i(t)$ for $t \geq T\text{-}1$ as follows:

$$I_i(t) \approx \sum_{u=T-T_{in}}^{t-1} \frac{\beta_i}{N_i} S_i(u) I_i(u) e^{-\gamma_i(t-u)}$$

$$= \sum_{u=T-T_{in}}^{t-1} \Delta I_i(u+1) e^{-\gamma_i(t-u)}, \tag{9}$$

where $\Delta I_i(u+1)$ represent the new infections at time step $u+1$, $\beta_i\, S_i(u) I_i(u)/ N_i$. Based on the relationships derived from Equations (6) and (7), $\Delta I_i(u+1)$ for $u \in [T\text{-}T_{in}, T\text{-}2]$ can be calculated as $(x_{i,u+2} - x_{i,u+1})/ \gamma_i + x_{i,u+1}$. In light of the above findings, at each time step $t \in [T, T+T_{out}]$ in region $i$, we update the states according to the following:

$$S_i(t) = S_i(t-1) - \frac{\beta_i}{N_i} S_i(t-1) I_i(t-1),$$

$$I_i(t) = \sum_{u=T-T_{in}}^{t-1} \Delta I_i(u+1) e^{-\gamma_i(t-u)},$$

$$R_i(t) = R_i(t-1) + \gamma_i I_i(t-1), \tag{10}$$

where the initial values at time step $T\text{-}1$ in each compartment are given by $I_i(T-1) = \sum_{u=T-T_{in}}^{T-2} \Delta I_i(u+1) e^{-\gamma_i(T-1-u)}$, $R_i(T-1) = \sum_{u}^{T-1} x_{i,u}$, $S_i(T-1) = N_i - I_i(T-1) - R_i(T-1)$. By applying the estimates of $\beta_i$ and $\gamma_i$ obtained from the spatio-temporal neural network module into Equation (10), and iteratively updating each state, we obtain the forecasted number of new confirmed cases $\hat{Y}_{i,t} \in \mathbb{R}$ at time step $t \in [T+1, T+T_{out}]$ in region $i$ as follows:

$$\hat{Y}_{i,t} = \gamma_i I_i(t-1) \tag{11}$$

**Algorithm 1** presents the pseudocode illustrating the flow leading to the prediction output.

Algorithm 1. PISID algorithm. **Spatio-temporal neural network module:**
1. $H_T \leftarrow$ Fully Connected Embedding (Input : $X_{T-T_{in}+1:T}$)
2. $E \leftarrow$ Spatial Embedding (Input : region $1, \ldots, M$)
3. $Z_T^{l+1} \leftarrow$ MLP based Encoding (Input : $H_T$, $E$)
4. $\beta, \gamma \leftarrow$ Fully Connected Regression (Input : $Z_T^{l+1}$)
**SIR module:**
5. $\{\Delta I_i(u)\}_{i=1,\ldots,M}^{u=T-T_{in}+1,\ldots,T-1} \leftarrow$ Calculate the historical new infections (Input : $X_{T-T_{in}+1:T}$, $\gamma$)

6. $\{S_i(T-1),\ I_i(T-1),\ R_i(T-1)\}_{i=1,\dots,M} \leftarrow$ Calculate the initial states (Input : $\{N_i\}_{i=1,\dots,M}$, $\{\Delta I_i(u)\}_{i=1,\dots,M}^{u=T-T_{in}+1,\dots,T-1}$, $\gamma$)

7. **for** each time step $t$ in $\{T,\dots,\ T+T_{out}\}$ **do**

8. $\{\hat{Y}_{i,t}\}_{i=1,\dots,M} \leftarrow$ Calculate the new confirmed cases (Input : $\{I_i(t-1)\}_{i=1,\dots,M}$, $\gamma$)

9. $\{\Delta I_i(t)\}_{i=1,\dots,M} \leftarrow$ Calculate the new infections (Input : $\{S_i(t-1),\ I_i(t-1),\ N_i\}_{i=1,\dots,M}$, $\beta$)

10. $\{S_i(t),\ I_i(t),\ R_i(t)\}_{i=1,\dots,M} \leftarrow$ Update each state (Input : $\{S_i(t-1), I_i(t-1), R_i(t-1), N_i\}_{i=1,\dots,M}$, $\{\Delta I_i(u)\}_{i=1,\dots,M}^{u=T-T_{in}+1,\dots,t}$, $\beta$, $\gamma$)

11. `return` $\{\hat{Y}_{i,t}\}_{i=1,\dots,M}^{t=T+1,\dots,T+T_{out}}$

## Objective function

We employ the Mean Absolute Error (MAE) as the loss function and train the model to capture the epidemic dynamics up to the target time step $T+T_{out}$ by minimizing the difference between the forecasted values $\hat{Y}_i = [\hat{Y}_{i,T+1}, \dots, \hat{Y}_{i,T+Tout}] \in R^{Tout}$ and the ground truth values $Y_i = [Y_{i,T+1}, \dots, Y_{i,T+Tout}] \in R^{Tout}$ for each region $i$. The objective function to be minimized is defined as:

$$\mathcal{L}(\Theta) = \frac{1}{M}\sum_{i=1}^{M}|\hat{Y}_i - Y_i| \tag{12}$$

where $\Theta$ denotes all learnable parameters, which are contained entirely within the spatio-temporal neural network module.

## Experimental study

### Datasets

To conduct our computational experiments, we use two publicly available COVID-19 datasets from Japan and the US, each recording the number of daily new confirmed cases:

• Japan: This dataset is collected from the Ministry of Health, Labour and Welfare [35] and contains the number of daily new confirmed cases for each of the 47 prefectures from January 16, 2020, to May 8, 2023. Population data for each prefecture are obtained from the Japan LIVE Dashboard [36].

• US: This dataset is sourced from the Johns Hopkins Coronavirus Resource Center [37] and includes the number of daily new confirmed cases for each of the 51 states from January 22, 2020, to March 9, 2023.

### Baselines

We compare the proposed PISID model with both traditional mathematical models (SIR, ARMA, GAR) and deep learning models (RNN, DCRNN, LSTNet, STGCN, GWNet, ColaGNN, FourierGNN, STID).

• SIR: The SIR model [5] is a classical compartmental model based on differential equations, widely used in epidemiology. We optimize the model parameters directly using historical data for each region.

• ARMA: ARMA is a fundamental statistical model for time series forecasting, which makes linear predictions based on past values and stochastic noise.

• GAR: GAR is an autoregressive model that incorporates inter-regional influence structures and is commonly used to model global economic systems.

• RNN: RNN [38] is a basic neural architecture for sequence modeling, which propagates information recursively from one time step to the next.

• DCRNN: DCRNN [39] is a spatio-temporal deep learning model that captures spatial dependencies via diffusion convolution and temporal dynamics via gated recurrent units.

 

- LSTNet: LSTNet [40] is a multivariate time series forecasting model that captures both short-term and long-term dependencies using a combination of CNN and RNN, and incorporates an autoregressive component to handle input scale variations.

- STGCN: STGCN [41] extracts spatial features using graph convolution and temporal features using gated temporal convolution.

- GWNet: GWNet [29] is a spatio-temporal deep learning model that adaptively learns the graph structure and captures spatio-temporal dependencies by combining graph convolution with dilated casual convolution.

- ColaGNN: ColaGNN [30] is an epidemic forecasting model that dynamically models spatial influence using a location-aware attention mechanism and captures local temporal patterns at multiple granularities using dilated convolution.

- FourierGNN: FourierGNN [42] is an architecture for multivariate time series forecasting that models spatio-temporal dynamics in a unified framework using matrix multiplication of space-time fully connected graphs with Fourier Graph Operators in Fourier space.

- STID: STID [10] is a multivariate time series forecasting model that addresses indistinguishability in spatial and temporal dimensions by embedding spatial and temporal identities through learnable matrices.

## Experimental setting

We evaluated our model under two forecasting scenarios: short-term and long-term. Both the input history length $T_{in}$ and the prediction horizon $T_{out}$ were set to either 14 or 28. This means the model predicts the number of new confirmed cases 14 or 28 days ahead using the past 14 or 28 days of data. The original daily case counts are heavily influenced by weekly seasonality, primarily due to the reporting practices of local governments and medical institutions—for example, a decrease in reports on weekends when many medical facilities are closed. To remove this seasonality, which is unrelated to actual infection trends, we applied a 7-day moving average as a preprocessing step. Additionally, because the dynamics of infection spread vary significantly depending on the dominant virus strain, we divided the dataset into two periods: one during which the Delta variant was dominant (Japan: 2020/01/22~2021/12/31, US: 2020/01/29~2021/11/30), and another during which the Omicron variant was dominant (Japan: 2022/01/01~2023/05/08, US: 2021/12/01~2023/03/09). Each dataset was split into training, validation, and test sets in a 6:1:3 ratio. Input data were normalized using the mean and standard deviation of the training set. The embedding dimension $D$ was set to 32, and the number of MLP layers in the encoder $L$ was set to 3. The number of model parameters to be trained was approximately 27K. We used a batch size of 32 and trained the model for up to 300 epochs, with early stopping triggered if validation performance does not improve for 20 consecutive epochs. Curriculum learning [43] was employed, gradually increasing the prediction horizon from 1 to $T_{out}$ by one time step every two epochs. We used the Adam optimizer with an initial learning rate of 0.001 and a weight decay of 1e-8. All experiments were conducted using PyTorch on a server with an NVIDIA A100 GPU. The code for PISID is available at https://github.com/satoki-fujita/PISID.

To evaluate predictive performance, we used the following metrics: Mean Absolute Error (MAE), Root Mean Square Error (RMSE), Mean Absolute Percentage Error (MAPE), Relative Absolute Error (RAE), Pearson Correlation Coefficient (PCC), and Concordance Correlation Coefficient (CCC). Lower values of MAE, RMSE, MAPE, and RAE, and higher values of PCC and CCC indicate better performance. These metrics are defined as follows:

$$\text{MAE} = \frac{1}{M}\sum_{i=1}^{M}|\hat{y}_i - y_i|, \tag{13}$$

$$\text{RMSE} = \sqrt{\frac{1}{M}\sum_{i=1}^{M}(\hat{y}_i - y_i)^2}, \tag{14}$$

$$\text{MAPE} = \frac{1}{M}\sum_{i=1}^{M}|\frac{\hat{y}_i - y_i}{y_i}|, \tag{15}$$

$$\text{RAE} = \frac{\sum_{i=1}^{M}|\hat{y}_i - y_i|}{\sum_{i=1}^{M}|\bar{y} - y_i|}, \tag{16}$$

$$\text{PCC} = \frac{\sum_{i=1}^{M}(\hat{y}_i - \bar{\hat{y}})(y_i - \bar{y})}{\sqrt{\sum_{i=1}^{M}\left(\hat{y}_i - \bar{\hat{y}}\right)^2}\sqrt{\sum_{i=1}^{M}\left(y_i - \bar{y}\right)^2}}, \tag{17}$$

$$\text{CCC} = \frac{2\rho\sigma_{\hat{y}}\sigma_y}{\sigma_{\hat{y}}^2 + \sigma_y^2 + \left(\bar{\hat{y}} - \bar{y}\right)^2}, \tag{18}$$

where $y_i$ denotes the observed value in region $i$, $\hat{y}_i$ is the predicted value in region $i$, $\bar{y}$ and $\hat{y}$ are the means of the observations and predictions, $\sigma_Y$ and $\sigma_{\hat{y}}$ are their standard deviations, and $\rho$ is the correlation coefficient between the observations and predictions.

### Prediction performance

We evaluated the predictive performance of each model on the test set. Each model was trained five times with different random initializations, and we report the mean and standard deviation of the evaluation metrics. The performance results for all models on the Japan dataset are presented in Table 1, and those for the US dataset are shown in Table 2. Across all datasets, PISID demonstrates consistent and competitive performance. In fact, in every case, it achieves either the best or the second-best MAE compared to other baseline models. On the Japan dataset (2020/01/22～2021/12/31), GWNet shows relatively strong performance, while on the Japan dataset (2022/01/01～2023/05/08), SIR performs comparatively well. However, the models do not exhibit notable performance when tested on the opposite time period, suggesting limited generalizability. These findings imply that the effectiveness of the models may be contingent upon the characteristics of the dominant epidemic dynamics in the time and place of application. During the Delta variant dominant period in Japan (2020/01/22–2021/12/31), government interventions such as mobility restrictions and limitations on restaurant operating hours were implemented periodically, which led to frequent changes in infection dynamics, resulting in relatively strong non-stationarity. Under such conditions, the adaptive nature of GWNet, which flexibly captures spatiotemporal dependencies, likely contributed to its effective performance. Meanwhile, during the Omicron variant dominant period in Japan (2022/01/01–2023/05/08), fewer abrupt interventions aimed at controlling human contact were implemented, allowing the epidemic dynamics to more closely follow the inherent epidemiological characteristics of the disease. Accordingly, the SIR model, grounded in epidemiological domain knowledge, is considered to have performed relatively well. GWNet did not exhibit distinctly superior performance during the period, potentially due to the complexity introduced by its graph structure learning mechanism which may have caused the model to overfit to spurious trends. In contrast, STID, which utilizes a straightforward neural network architecture devoid of graph-based components, achieved more favorable results. PISID, which integrates the STID architecture with the SIR model, have been capable of handling both scenarios where complex spatiotemporal patterns predominate and those where epidemiologically specific dynamics are dominant, without experiencing significant performance degradation. Furthermore, PISID consistently maintained its performance regardless of the forecast horizon. Other neural network models search a vast representational space for epidemic dynamics that best fit the training data without any guidelines based on epidemiological knowledge, which

**Table 1. Prediction performance on the Japan dataset.**

| | Dataset Period: 2020/01/22~2021/12/31 | | | | | | | | | | | |
|---|---|---|---|---|---|---|---|---|---|---|---|---|
| | *Tin, Tout* = 14 | | | | | | *Tin, Tout* = 28 | | | | | |
| Model | MAE | RMSE | MAPE | RAE | PCC | CCC | MAE | RMSE | MAPE | RAE | PCC | CCC |
| SIR | 80.311 (-) | 312.229 (-) | 5.606 (-) | 0.561 (-) | 0.896 (-) | 0.793 (-) | 296.314 (-) | 1080.897 (-) | 8.344 (-) | 2.006 (-) | 0.591 (-) | 0.312 (-) |
| ARMA | 58.049 (1.592) | 205.957 (4.790) | 1.236 (0.033) | 0.406 (0.011) | 0.856 (0.017) | 0.762 (0.011) | 89.635 (2.073) | 308.028 (7.531) | 3.139 (0.051) | 0.607 (0.014) | 0.572 (0.044) | 0.411 (0.024) |
| GAR | 56.818 (0.380) | 186.598 (0.719) | 1.522 (0.021) | 0.397 (0.003) | 0.857 (0.002) | 0.830 (0.003) | 98.372 (0.295) | 315.117 (1.594) | 5.299 (0.077) | 0.666 (0.002) | 0.516 (0.006) | 0.385 (0.012) |
| RNN | 54.569 (1.539) | 203.979 (8.230) | 1.328 (0.140) | 0.381 (0.011) | 0.848 (0.006) | 0.769 (0.028) | 98.587 (0.242) | 325.258 (1.141) | 4.447 (0.092) | 0.667 (0.002) | 0.517 (0.003) | 0.277 (0.008) |
| DCRNN | 67.716 (2.092) | 250.461 (6.405) | 1.592 (0.181) | 0.473 (0.015) | 0.804 (0.013) | 0.593 (0.029) | 92.210 (3.064) | 323.985 (4.748) | 3.978 (0.816) | 0.624 (0.021) | 0.583 (0.048) | 0.278 (0.028) |
| LSTNet | 68.666 (5.404) | 255.767 (20.163) | 1.484 (0.063) | 0.480 (0.038) | 0.808 (0.051) | 0.567 (0.088) | 83.450 (1.876) | 311.655 (13.161) | <u>1.844</u> (0.261) | 0.565 (0.013) | 0.788 (0.024) | 0.329 (0.067) |
| STGCN | 65.534 (6.341) | 274.521 (28.919) | 1.412 (0.413) | 0.458 (0.044) | 0.830 (0.066) | 0.465 (0.127) | 84.273 (1.566) | 319.103 (4.635) | 2.798 (0.821) | 0.571 (0.011) | 0.833 (0.019) | 0.288 (0.023) |
| GWNet | 54.688 (4.845) | 224.647 (17.790) | **0.894** (0.178) | 0.382 (0.034) | 0.914 (0.038) | 0.672 (0.060) | <u>74.423</u> (2.691) | 275.644 (16.280) | **1.245** (0.156) | <u>0.504</u> (0.018) | **0.893** (0.034) | 0.491 (0.069) |
| Cola-GNN | 67.381 (5.928) | 269.036 (22.092) | 1.069 (0.106) | 0.471 (0.041) | 0.790 (0.127) | 0.516 (0.080) | 79.307 (7.024) | 282.608 (25.451) | 2.210 (0.409) | 0.537 (0.048) | <u>0.842</u> (0.091) | 0.465 (0.120) |
| Fouri-erGNN | 69.358 (9.822) | 254.482 (53.841) | 1.521 (0.168) | 0.485 (0.069) | 0.837 (0.012) | 0.760 (0.091) | 96.273 (3.141) | 334.089 (5.624) | 3.804 (0.374) | 0.652 (0.021) | 0.467 (0.017) | 0.255 (0.063) |
| STID | <u>45.690</u> (1.960) | <u>172.350</u> (7.326) | <u>1.025</u> (0.033) | <u>0.319</u> (0.014) | <u>0.923</u> (0.013) | <u>0.836</u> (0.017) | 76.448 (1.998) | <u>274.194</u> (12.139) | 2.350 (0.193) | 0.518 (0.014) | 0.801 (0.020) | <u>0.511</u> (0.058) |
| PISID | **41.814** (1.399) | **142.584** (5.825) | 1.201 (0.081) | **0.292** (0.010) | **0.926** (0.007) | **0.904** (0.012) | **69.721** (3.168) | **242.307** (10.188) | 2.109 (0.306) | **0.472** (0.021) | 0.819 (0.040) | **0.653** (0.036) |
| | Dataset Period: 2022/01/01~2023/05/08 | | | | | | | | | | | |
| | $T_{in}$, $T_{out}$ = 14 | | | | | | $T_{in}$, $T_{out}$ = 28 | | | | | |
| Model | MAE | RMSE | MAPE | RAE | PCC | CCC | MAE | RMSE | MAPE | RAE | PCC | CCC |
| SIR | 377.464 (-) | 1074.607 (-) | <u>0.356</u> (-) | 0.327 (-) | 0.897 (-) | 0.878 (-) | 594.251 (-) | 1469.205 (-) | **0.733** (-) | 0.588 (-) | **0.854** (-) | **0.781** (-) |
| ARMA | 372.730 (2.181) | <u>803.702</u> (6.046) | 0.412 (0.006) | 0.323 (0.002) | 0.911 (0.002) | <u>0.903</u> (0.001) | <u>553.148</u> (2.382) | <u>1056.272</u> (7.149) | 1.087 (0.013) | <u>0.547</u> (0.002) | 0.797 (0.003) | 0.777 (0.003) |
| GAR | 357.219 (3.640) | 824.248 (21.105) | 0.391 (0.014) | <u>0.310</u> (0.003) | 0.906 (0.005) | 0.898 (0.005) | 568.604 (46.105) | 1119.846 (48.692) | 0.944 (0.266) | 0.562 (0.046) | 0.771 (0.022) | 0.743 (0.045) |
| RNN | 368.593 (5.979) | 857.042 (43.048) | 0.456 (0.023) | 0.320 (0.005) | 0.900 (0.010) | 0.896 (0.009) | 602.137 (31.846) | 1216.052 (12.487) | 1.101 (0.121) | 0.596 (0.032) | 0.719 (0.006) | 0.671 (0.014) |
| DCRNN | 537.941 (25.521) | 1029.986 (17.778) | 1.044 (0.129) | 0.466 (0.022) | 0.863 (0.003) | 0.813 (0.011) | 659.606 (39.250) | 1279.289 (16.153) | 1.672 (0.210) | 0.653 (0.039) | 0.721 (0.017) | 0.567 (0.015) |
| LSTNet | 663.108 (169.656) | 1199.806 (284.719) | 1.363 (0.500) | 0.575 (0.147) | 0.771 (0.113) | 0.723 (0.127) | 1356.146 (172.509) | 2234.250 (394.957) | 5.398 (0.749) | 1.342 (0.171) | 0.271 (0.120) | 0.254 (0.115) |
| STGCN | 537.312 (45.562) | 1159.295 (158.659) | 1.262 (0.337) | 0.466 (0.040) | 0.888 (0.010) | 0.854 (0.016) | 1392.695 (241.037) | 2834.379 (504.246) | 5.650 (1.439) | 1.378 (0.238) | 0.413 (0.106) | 0.338 (0.102) |
| GWNet | 405.937 (16.229) | 893.528 (28.986) | 0.690 (0.120) | 0.352 (0.014) | 0.911 (0.005) | 0.903 (0.004) | 852.799 (175.577) | 1500.963 (375.509) | 3.886 (1.474) | 0.844 (0.174) | 0.615 (0.154) | 0.574 (0.153) |
| Cola-GNN | 509.714 (55.936) | 953.039 (124.566) | 1.448 (0.359) | 0.442 (0.049) | 0.886 (0.029) | 0.841 (0.062) | 944.550 (234.498) | 2136.036 (917.586) | 3.704 (1.921) | 0.934 (0.232) | 0.392 (0.253) | 0.346 (0.230) |
| Fouri-erGNN | 377.684 (26.320) | 851.524 (55.036) | 0.553 (0.149) | 0.327 (0.023) | 0.905 (0.012) | 0.894 (0.007) | 614.848 (37.821) | 1218.852 (38.813) | 1.275 (0.233) | 0.608 (0.037) | 0.740 (0.022) | 0.633 (0.040) |
| STID | <u>361.919</u> (10.562) | 807.538 (38.050) | 0.421 (0.048) | 0.314 (0.009) | <u>0.917</u> (0.003) | **0.914** (0.004) | 579.034 (26.859) | **1053.597** (13.914) | 1.279 (0.188) | 0.573 (0.027) | <u>0.802</u> (0.006) | <u>0.778</u> (0.014) |

*(Continued)*

**Table 1.** (Continued)

| Model | Dataset Period: 2020/01/22~2021/12/31 | | | | | | | | | | | |
| | Tin, Tout = 14 | | | | | | Tin, Tout = 28 | | | | | |
| | MAE | RMSE | MAPE | RAE | PCC | CCC | MAE | RMSE | MAPE | RAE | PCC | CCC |
| PISID | **331.010** (11.551) | **785.949** (54.061) | **0.298** (0.056) | **0.287** (0.010) | **0.927** (0.003) | 0.900 (0.020) | **549.221** (61.100) | 1160.686 (123.978) | <u>0.750</u> (0.237) | **0.543** (0.060) | 0.748 (0.068) | 0.709 (0.076) |

The performance values are mean (std). The bold values indicate the best results, the underlined values indicate the second-best results.

increases the risk of overfitting and can lead to a more pronounced performance degradation when transitioning from short-term to long-term forecasts. For example, on the US dataset (2021/12/01~2023/03/09), GWNet performs well for 14-day-ahead forecasts but suffers a more significant drop in accuracy for 28-day-ahead forecasts compared to PISID. Even STID, which demonstrates competitive performance on other datasets, exhibits a similarly significant deterioration. Given the sparse and noisy nature of infectious disease data, incorporating epidemiological domain knowledge—as done in PISID—contributes to more stable and reliable predictions.

Fig 2 visualizes the 28-day-ahead forecasts produced by PISID and representative baseline models on the test set for Tokyo and New York, alongside the actual observed values. The right column of the figure reveals that ColaGNN's forecasts are markedly unstable, with pronounced divergence from the ground truth values, likely caused by overfitting due to the attention mechanism used for graph structure learning, which leads to excessive model size. In contrast, PISID produces relatively stable forecasts; however, like other models, it struggles to capture the real-time dynamics of infection spread. A 28-day forecasting horizon is long enough for the epidemic distribution to change, and it is possible that there is a lack of external data capable of capturing such changes. Especially, abrupt outbreaks are likely driven by some kind of external intervention, making it challenging to predict their onset accurately based solely on historical confirmed case data.

In addition to the predictive performance, we also evaluated the training efficiency of the neural network models. Table 3 presents the training time per epoch for each model on each dataset. While PISID requires more training time than STID due to the inclusion of the SIR module that performs iterative processing, it is more computationally efficient than more complex models that adaptively learn graph structures, such as GWNet and ColaGNN.

To verify the effectiveness of the fully connected neural network-based encoder with a spatial embedding matrix in the neural network module, we also compared performance when replacing the encoder with alternative architectures. We employed several commonly used models for time-series tasks as encoders, including RNN [38], which uses recurrent architectures to process time-series information, TCN [44], which employs convolutional architectures for sequence modeling, and Transformer [45], which utilizes an attention mechanism to capture long-range dependencies in sequences. In addition, we evaluated GWNet [29], a spatio-temporal model that adaptively learns graph structures, as the encoder, and also assessed a variant of our model without the spatial embedding matrix to investigate the contribution of spatial embeddings to performance. In all cases, the encoded information was decoded into epidemiological parameters via a fully connected layer and passed to the subsequent SIR module. Table 4 presents the MAE and RMSE scores for predictions made by models using each encoder architecture across the datasets. Among the encoder architectures that do not explicitly incorporate spatial information—namely RNN, TCN, Transformer, and MLP w/o SID—MLP w/o SID demonstrates competitive performance compared to other sequence-specialized encoders, suggesting that MLP-based architectures can effectively capture temporal dependencies. Furthermore, MLP w/ SID, which incorporates a spatial embedding matrix, achieves the best performance among all encoder architectures, including GWNet that perform adaptive graph structure learning. This underscores the efficacy of handling spatial dependencies using a simple embedding-based approach.

**Table 2. Prediction performance on the US dataset.**

| | Dataset Period: 2020/01/29~2021/11/30 | | | | | | | | | | | |
|---|---|---|---|---|---|---|---|---|---|---|---|---|
| | *Tin, Tout* = 14 | | | | | | *Tin, Tout* = 28 | | | | | |
| Model | MAE | RMSE | MAPE | RAE | PCC | CCC | MAE | RMSE | MAPE | RAE | PCC | CCC |
| SIR | 510.437 (-) | 1284.009 (-) | 0.434 (-) | 0.351 (-) | 0.920 (-) | 0.894 (-) | 1145.956 (-) | 3250.338 (-) | 0.763 (-) | 0.772 (-) | 0.792 (-) | 0.640 (-) |
| ARMA | 514.403 (67.744) | 1066.408 (168.824) | 0.487 (0.057) | 0.354 (0.047) | 0.903 (0.033) | 0.883 (0.040) | 957.729 (17.773) | 2058.915 (41.378) | 0.946 (0.067) | 0.645 (0.012) | 0.606 (0.029) | 0.508 (0.020) |
| GAR | 423.269 (9.377) | 951.196 (24.588) | 0.374 (0.007) | 0.291 (0.006) | 0.937 (0.002) | 0.907 (0.006) | 905.968 (97.416) | 1974.908 (174.796) | 0.866 (0.043) | 0.610 (0.066) | 0.648 (0.083) | 0.533 (0.079) |
| RNN | 436.910 (8.840) | 987.069 (26.241) | 0.363 (0.007) | 0.301 (0.006) | 0.932 (0.005) | 0.899 (0.008) | 847.424 (13.532) | 1872.446 (25.903) | 0.888 (0.024) | 0.571 (0.009) | 0.699 (0.011) | 0.570 (0.018) |
| DCRNN | 533.987 (23.286) | 1152.236 (65.860) | 0.485 (0.033) | 0.368 (0.016) | 0.891 (0.014) | 0.862 (0.019) | 855.196 (27.689) | 1821.558 (51.977) | 1.007 (0.094) | 0.576 (0.019) | 0.697 (0.020) | 0.616 (0.034) |
| LSTNet | 721.123 (60.294) | 1543.540 (107.111) | 0.671 (0.068) | 0.497 (0.042) | 0.781 (0.034) | 0.761 (0.025) | 1070.836 (66.232) | 2176.730 (92.266) | 1.320 (0.242) | 0.721 (0.045) | 0.531 (0.039) | 0.484 (0.025) |
| STGCN | 696.698 (34.219) | 1512.967 (112.930) | 0.543 (0.035) | 0.480 (0.024) | 0.804 (0.029) | 0.791 (0.021) | 887.266 (37.381) | 1977.124 (91.256) | 0.797 (0.085) | 0.597 (0.025) | 0.658 (0.047) | 0.583 (0.012) |
| GWNet | 437.572 (50.186) | 974.651 (120.673) | 0.381 (0.028) | 0.301 (0.035) | 0.922 (0.019) | 0.912 (0.018) | 896.924 (109.380) | 1918.395 (206.878) | 0.858 (0.055) | 0.604 (0.074) | 0.693 (0.045) | 0.652 (0.043) |
| Cola-GNN | 614.346 (120.075) | 1413.435 (314.927) | 0.561 (0.133) | 0.423 (0.083) | 0.864 (0.045) | 0.815 (0.100) | 842.428 (81.433) | 1801.807 (127.591) | 0.910 (0.168) | 0.567 (0.055) | 0.740 (0.058) | 0.638 (0.080) |
| Fouri-erGNN | 465.345 (27.814) | 1036.614 (112.079) | 0.419 (0.023) | 0.320 (0.019) | 0.930 (0.013) | 0.890 (0.040) | 853.659 (72.122) | 1891.898 (171.132) | 0.982 (0.181) | 0.575 (0.049) | 0.746 (0.041) | 0.547 (0.105) |
| STID | **388.584** (15.023) | **895.060** (50.051) | **0.337** (0.010) | **0.268** (0.010) | **0.945** (0.010) | **0.919** (0.009) | **661.425** (10.111) | **1458.985** (46.441) | <u>0.653</u> (0.045) | **0.445** (0.007) | **0.856** (0.010) | <u>0.759</u> (0.024) |
| PISID | <u>408.179</u> (8.363) | <u>939.972</u> (37.409) | <u>0.339</u> (0.008) | **0.281** (0.006) | <u>0.940</u> (0.003) | 0.910 (0.009) | <u>714.680</u> (21.899) | <u>1495.119</u> (58.309) | **0.577** (0.010) | <u>0.481</u> (0.015) | <u>0.832</u> (0.011) | **0.767** (0.026) |
| | Dataset Period: 2021/12/01~2023/03/09 | | | | | | | | | | | |
| | *T_in, T_out* = 14 | | | | | | *T_in, T_out* = 28 | | | | | |
| Model | MAE | RMSE | MAPE | RAE | PCC | CCC | MAE | RMSE | MAPE | RAE | PCC | CCC |
| SIR | 573.112 (-) | 1554.133 (-) | **0.891** (-) | 0.709 (-) | 0.654 (-) | 0.592 (-) | 827.365 (-) | 2786.604 (-) | 2.695 (-) | 1.010 (-) | 0.470 (-) | 0.336 (-) |
| ARMA | 764.790 (476.672) | 9303.936 (11850.750) | 3.162 (0.342) | 0.947 (0.590) | 0.291 (0.227) | 0.248 (0.253) | 3350.136 (3113.137) | 83261.520 (111957.984) | 5.654 (1.820) | 4.088 (3.799) | 0.044 (0.019) | 0.006 (0.006) |
| GAR | 523.325 (173.168) | 3020.446 (2602.908) | 1.982 (0.293) | 0.648 (0.214) | 0.524 (0.258) | 0.454 (0.305) | 564.081 (56.930) | 1594.138 (644.614) | 2.709 (0.127) | 0.688 (0.069) | 0.419 (0.115) | 0.358 (0.089) |
| RNN | 335.160 (16.795) | <u>742.980</u> (28.331) | 1.608 (0.144) | 0.415 (0.021) | 0.845 (0.007) | 0.790 (0.021) | 423.198 (11.253) | 938.667 (20.483) | 2.945 (0.117) | 0.516 (0.014) | 0.763 (0.004) | 0.613 (0.026) |
| DCRNN | 414.063 (41.026) | 841.853 (59.833) | 2.360 (0.468) | 0.512 (0.051) | 0.790 (0.027) | 0.744 (0.095) | 608.135 (55.094) | 1061.608 (82.243) | 4.302 (0.735) | 0.742 (0.067) | 0.675 (0.052) | 0.496 (0.190) |
| LSTNet | 468.734 (66.018) | 1284.823 (641.536) | 3.239 (0.935) | 0.580 (0.082) | 0.755 (0.161) | 0.690 (0.181) | 775.788 (140.892) | 2277.587 (386.156) | 5.294 (0.438) | 0.947 (0.172) | 0.588 (0.130) | 0.451 (0.106) |
| STGCN | 383.939 (11.253) | 818.942 (25.384) | 2.096 (0.115) | 0.475 (0.014) | 0.803 (0.018) | 0.748 (0.017) | <u>401.808</u> (14.354) | <u>866.058</u> (30.985) | 3.033 (0.347) | <u>0.490</u> (0.018) | <u>0.777</u> (0.026) | <u>0.761</u> (0.035) |
| GWNet | **298.809** (8.078) | **664.301** (23.912) | 1.120 (0.152) | **0.370** (0.010) | **0.865** (0.010) | **0.861** (0.009) | 428.229 (17.755) | 928.569 (60.528) | 2.926 (0.514) | 0.523 (0.022) | 0.754 (0.019) | 0.732 (0.037) |
| Cola-GNN | 330.912 (21.375) | 764.109 (61.557) | 1.201 (0.203) | 0.410 (0.026) | 0.828 (0.020) | 0.819 (0.026) | 489.655 (57.388) | 969.138 (102.001) | 3.310 (0.270) | 0.598 (0.070) | 0.769 (0.012) | 0.739 (0.037) |
| Fouri-erGNN | 388.369 (26.953) | 1075.503 (185.613) | 1.433 (0.331) | 0.481 (0.033) | 0.668 (0.085) | 0.639 (0.070) | 552.500 (56.676) | 1468.967 (589.867) | <u>2.516</u> (0.457) | 0.674 (0.069) | 0.481 (0.155) | 0.383 (0.127) |
| STID | 342.879 (16.679) | 836.780 (44.723) | 1.210 (0.039) | 0.424 (0.021) | 0.826 (0.013) | 0.819 (0.015) | 472.337 (16.097) | 1216.151 (119.765) | 3.391 (0.409) | 0.576 (0.020) | 0.612 (0.093) | 0.604 (0.085) |

*(Continued)*

**Table 2.** (Continued)

| Model | Dataset Period: 2020/01/29~2021/11/30 | | | | | | | | | | | |
| | Tin, Tout = 14 | | | | | | Tin, Tout = 28 | | | | | |
| | MAE | RMSE | MAPE | RAE | PCC | CCC | MAE | RMSE | MAPE | RAE | PCC | CCC |
| PISID | 326.918 (7.983) | 792.550 (37.998) | 1.004 (0.102) | 0.405 (0.010) | 0.828 (0.012) | 0.827 (0.013) | 370.447 (6.435) | 806.841 (26.431) | 2.409 (0.136) | 0.452 (0.008) | 0.811 (0.013) | 0.807 (0.012) |

The performance values are mean (std). The bold values indicate the best results, the underlined values indicate the second-best results.

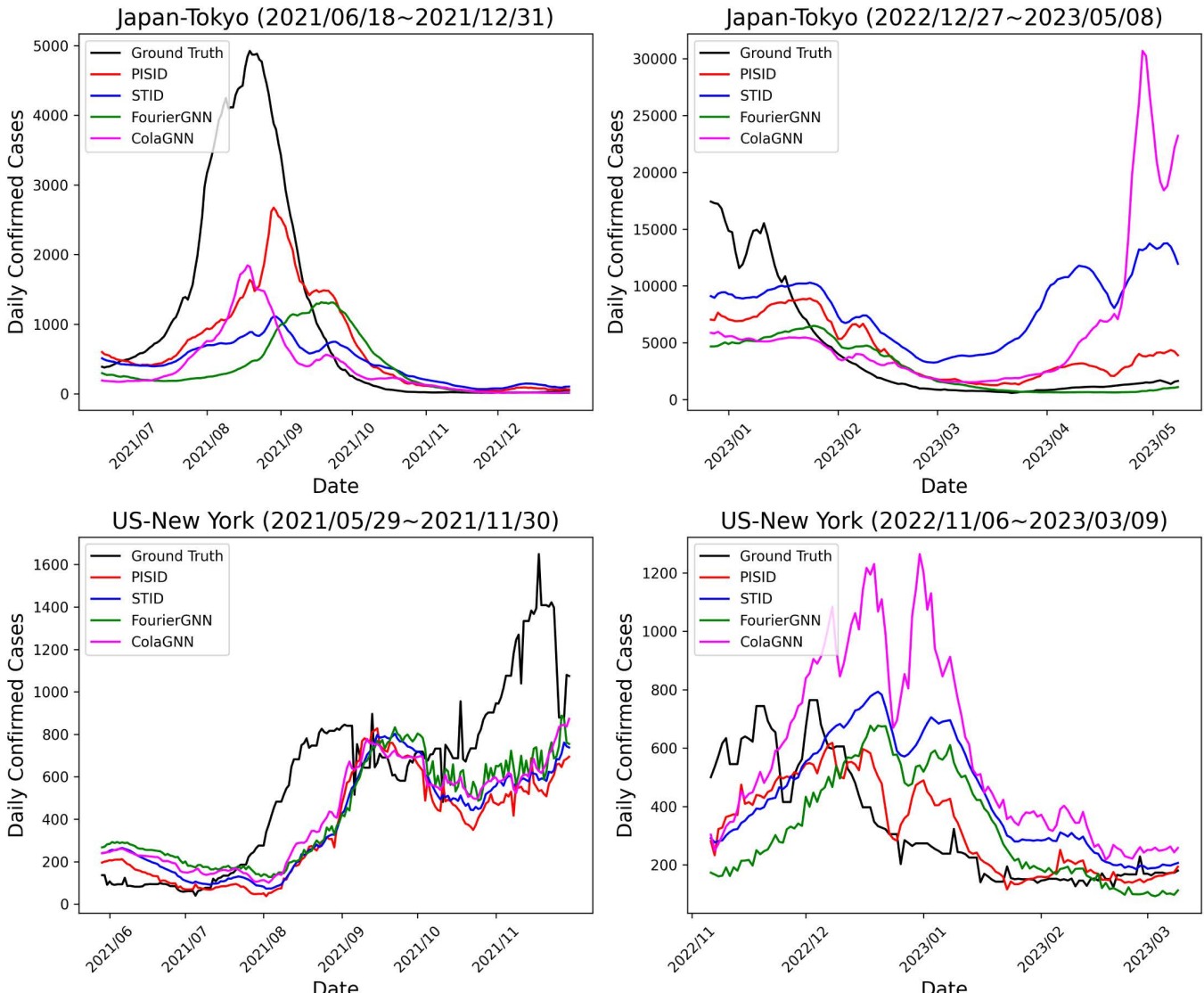

**Fig 2. Plot of the predicted confirmed cases 28-day-ahead in Tokyo and New York.**

**Table 3. Runtime on each dataset ($T_{in}$, $T_{out}$ = 28).**

| Dataset | Japan dataset | | US dataset | |
|---|---|---|---|---|
| | 2020/01/22～2021/12/31 | 2022/01/01～2023/05/08 | 2020/01/29～2021/11/30 | 2021/12/01～2023/03/09 |
| Model | Runtime (seconds/epoch) | | | |
| RNN | 0.04376 | 0.028365 | 0.057498 | 0.037098 |
| DCRNN | 1.616268 | 1.11854 | 1.475727 | 1.023883 |
| LSTNet | 0.060737 | 0.04237 | 0.059269 | 0.040026 |
| STGCN | 0.109183 | 0.075463 | 0.08844 | 0.0621 |
| GWNet | 5.408724 | 3.012791 | 4.06402 | 2.719374 |
| ColaGNN | 2.761671 | 1.841789 | 3.720598 | 1.906458 |
| FourierGNN | 0.097965 | 0.06824 | 0.077494 | 0.053875 |
| STID | 0.066253 | 0.037091 | 0.058131 | 0.033581 |
| PISID | 0.542497 | 0.37584 | 0.521152 | 0.3633 |

**Table 4. Prediction performance of models with different backbone encoder architectures across each dataset ($T_{in}$, $T_{out}$ = 28).**

| backbone encoder | Japan dataset | | | | US dataset | | | |
|---|---|---|---|---|---|---|---|---|
| | 2020/01/22～2021/12/31 | | 2022/01/01～2023/05/08 | | 2020/01/29～2021/11/30 | | 2021/12/01～2023/03/09 | |
| | MAE | RMSE | MAE | RMSE | MAE | RMSE | MAE | RMSE |
| RNN | 95.427 (5.637) | 292.646 (16.486) | 758.801 (533.859) | 1862.011 (1538.003) | 784.039 (38.427) | 1724.497 (99.907) | 465.061 (14.429) | 1016.348 (48.749) |
| TCN | 89.703 (7.053) | 284.813 (6.780) | 775.389 (584.968) | 1885.780 (1426.976) | 773.209 (20.584) | 1655.712 (39.563) | 456.034 (26.577) | 978.103 (35.039) |
| Trans-former | 94.871 (3.890) | 295.708 (1.107) | 634.091 (239.881) | 1290.531 (501.541) | 753.142 (21.311) | 1643.688 (52.059) | 439.512 (19.198) | 952.822 (32.240) |
| GWNet | 74.078 (4.391) | 248.201 (16.707) | 631.808 (274.681) | 1256.503 (413.829) | 840.253 (30.579) | 1784.920 (58.166) | 398.896 (5.915) | 843.079 (17.294) |
| MLP w/o SID | 82.339 (2.182) | 269.679 (8.83) | 620.335 (251.957) | 1373.702 (435.859) | 780.706 (49.844) | 1644.267 (88.948) | 451.672 (3.463) | 958.609 (13.084) |
| MLP w/ SID (*Ours*) | **69.721** (3.168) | **242.307** (10.188) | **549.221** (61.100) | **1160.686** (123.978) | **714.68** (21.899) | **1495.119** (58.309) | **370.447** (6.435) | **806.841** (26.431) |

The performance values are mean (std). The bold values indicate the best results.

We also examined the sensitivity of the hidden dimension $D$, which corresponds to the dimensionality of the temporal feature $H_T$ and the spatial feature $E$ embedded by the encoder. The dimension values were set to {8, 16, 32, 64, 128}, and evaluation results for each dataset are presented in Fig 3. When $D$ is too small, the embedded spatio-temporal representation becomes limited, resulting in degraded predictive performance. On the other hand, as observed in the results for the Delta strain epidemic data (Fig 3, left), an excessively large $D$ may lead to inferior performance due to overfitting. Therefore, selecting a well-balanced value for $D$ is recommended.

## Interpretability

Since the PISID model incorporates an SIR module, its predictions can be made interpretable through the parameters that govern the underlying dynamical system. We focus on the effective reproduction number $R_e(T)$, defined as $\beta/\gamma \cdot S(T)/N$, a widely used indicator of infectious disease transmissibility. To assess the interpretability of the model, we conducted a case study using this metric. In Fig 4, we use the PISID model configured for 28-day-ahead forecasting ($T_{out}$ = 28) and

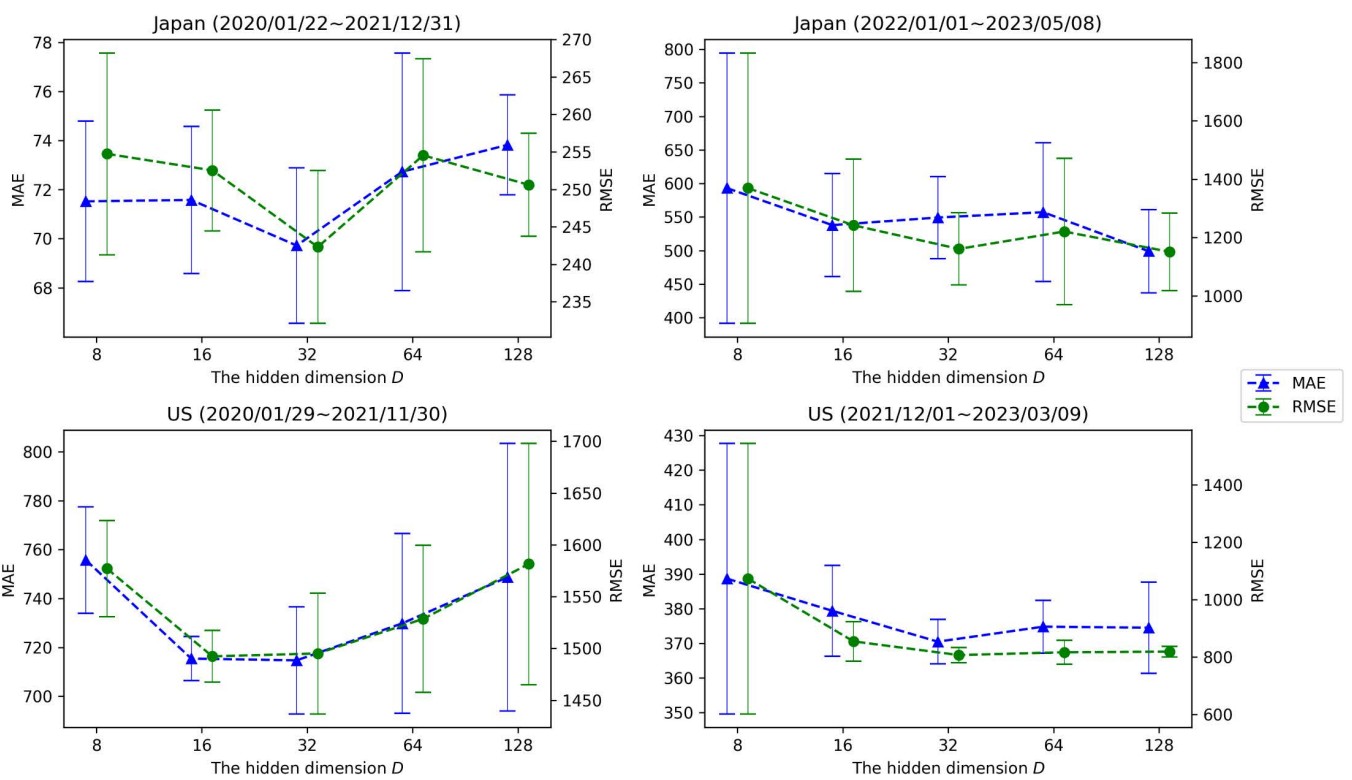

**Fig 3. Sensitivity analysis results of the hidden dimension *D* across each dataset (*Tin*, *Tout* = 28).**

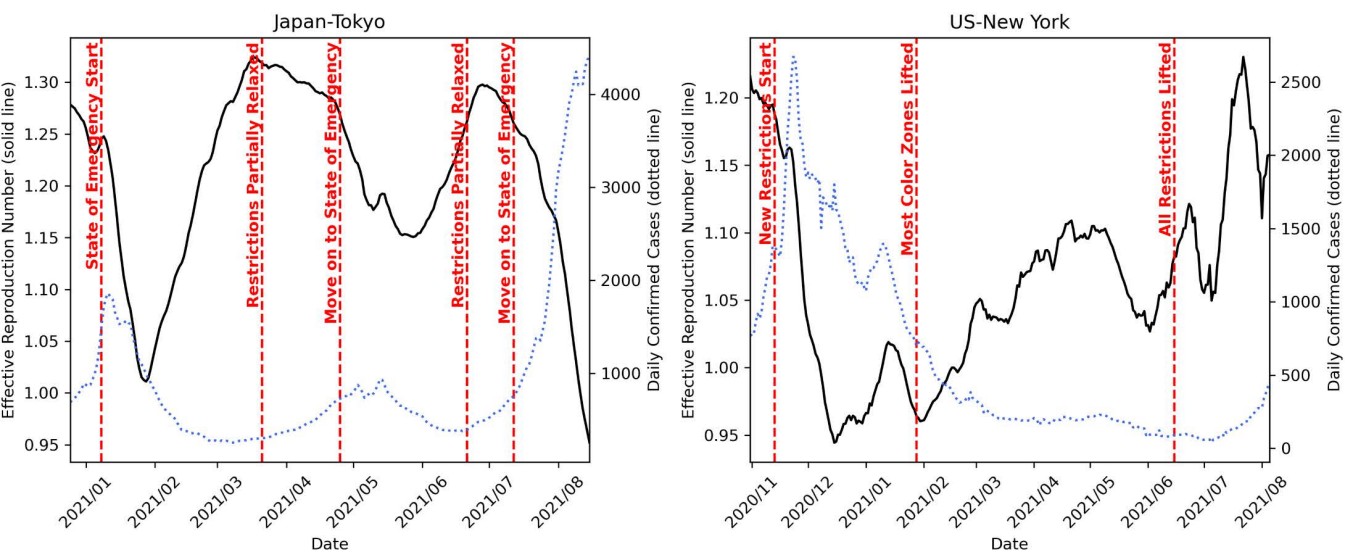

**Fig 4. Plot of the derived effective reproduction number with event labels in Tokyo and New York.**

plot the estimated $R_e$ values over time $T$ for specific periods in which significant COVID-19 response measures were implemented in the training dataset for Tokyo (Japan) and New York (US). The plot also includes major event labels along the timeline and the actual daily confirmed cases. In Tokyo, $R_e$ begins to decline sharply after the declarations of a state of emergency on 2021/01/08, 2021/04/25, and 2021/07/12. During each of these periods, residents were urged to stay home, and customer-facing establishments were requested to shorten operating hours. The behavior of $R_e$ appears to reflect the reduction in infection risk resulting from these externally enforced measures. Conversely, $R_e$ increases again around 2021/03/22 and 2021/06/21, when case numbers had declined and restrictions were partially lifted—suggesting a potential resurgence of infections following deregulation. Indeed, the number of newly confirmed cases began to increase following each of these points in time. In New York, $R_e$ drops significantly after 2020/11/13, when new restrictions were imposed on restaurants, bars, gyms, and private gatherings, falling below 1—a threshold often interpreted as indicating that the epidemic is under control. A decline in the number of newly confirmed cases can also be observed, as if mirroring this trend. $R_e$ begins to rise again on 2021/01/28, possibly reflecting the gradual easing of restrictions and the lifting of most "color zone" regulations. Between 2021/07~2021/08, just prior to the resurgence driven by the Delta variant, a notable increase is also observed. This trend may be associated with the full lifting of restrictions on 2021/06/15. These results suggest that $R_e$, as estimated by PISID, reflects real-world fluctuations in transmission dynamics in a relatively timely and interpretable manner. It can thus serve as a meaningful indicator for assessing the epidemic situation based on the model's internal epidemiological reasoning.

## Discussion

We proposed PISID, a simple infectious disease forecasting model that combines a fully connected neural network with an SIR module, and evaluated its performance using real-world COVID-19 case data from Japan and the US. Although PISID's predictions are grounded in the deterministic dynamics of the SIR model, it demonstrates competitive predictive performance compared to well-established neural network baselines. This highlights the importance of incorporating domain knowledge in infectious disease modeling. While neural networks can flexibly approximate complex functions through a large number of parameters, relying solely on noisy data—such as epidemic time series—can lead to overfitting and poor generalization. Embedding prior knowledge of epidemic dynamics into the model architecture, especially in scenarios where large-scale training data or external features are limited, can enhance generalization and robustness. The SIR module in PISID, though a simplified dynamical system representing average epidemic behavior in a population, maintains strong empirical performance without compromising the validity of its underlying principles. Moreover, it contributes to addressing the interpretability challenges often associated with neural networks. The parameters estimated by the SIR module can be interpreted as indicators of future transmissibility, offering practical value for outbreak risk management. For instance, an increase in the parameter value can serve as an early warning signal, enabling timely interventions such as contact tracing or resource allocation. Conversely, a decrease in the parameter may indicate a suitable time to relax restrictions. This level of interpretability is particularly important in the context of infectious diseases, which can have far-reaching societal, economic, and healthcare impacts, thereby enhancing the model's practical utility.

It is also noteworthy that the neural network component of PISID primarily consists of basic fully connected layers, without relying on graph structure learning. While recurrent or convolutional architectures are commonly used for sequence modeling due to their memory capabilities, our experimental results show that MLP-based structures are equally effective in capturing temporal dynamics. In fact, recent studies have reported that simple linear layers can outperform more complex architectures like Transformers [46], which also serves as the foundational architecture for large language models (LLMs), suggesting that simplicity should not be underestimated. The lightweight nature of MLP also enables efficient training without excessive computational overhead. Spatial dependencies are captured using a spatial embedding matrix, avoiding the complexity of graph structure learning employed in many spatio-temporal models. Overall, the architecture

of PISID is straightforward and interpretable, yet it effectively encodes both spatial and temporal information, achieving performance comparable to more complex models.

There are, however, several limitations and directions for future work. First, this study focuses on forecasting future confirmed cases using only past case data as input. Since epidemic dynamics are influenced by various external factors—such as cluster outbreaks, viral mutations, new treatments or vaccines, and government interventions—incorporating additional external data could improve predictive accuracy.

Second, since the proposed method generates forecasts based on the SIR equations, it performs well in predicting stationary epidemic patterns but struggles to respond sensitively to sudden trend shifts. In infectious diseases such as COVID-19, distribution characteristics can change abruptly due to mutations in virus strains or shifts in human behavior. In situations where such non-stationarity is pronounced, predictive performance becomes limited. In our experiments, the dataset was pre-divided based on a known major change point—specifically, the shift from the Delta to the Omicron variant—allowing the method to be evaluated under relatively stationary conditions. Addressing prediction under broader, potentially more non-stationary scenarios remains an important future challenge. It is necessary either to attempt predictions within each stationary pattern interval, in conjunction with detecting change points where the distribution shifts, or to develop a model that incorporates new mechanisms capable of responding sensitively to non-stationary epidemic patterns. In addition, the use of the aforementioned external data associated with the dynamics of non-stationary epidemic patterns, is expected to be essential for detecting such patterns.

Third, our experiments are limited to COVID-19 data. Further research is needed to assess the model's applicability to other infectious diseases, such as influenza. Depending on the disease, alternative compartmental models (e.g., SIS) may better represent the transmission process. Extending PISID to support such model variants could further enhance its generalizability. Additionally, for infectious diseases with strong periodicity, it may be necessary to develop models that account for periodic patterns, such as C-GRU [47].

## Conclusion

In this paper, we proposed PISID, a novel model for epidemic forecasting across multiple regions. PISID combines an SIR module—based on an infectious disease-specific dynamical system—with a simple neural network module composed of fully connected layers. The model requires only historical confirmed case data as input, making it broadly applicable. While complex models that incorporate graph structure learning can sometimes suffer from overfitting and limited interpretability, PISID is designed to follow an exponential trajectory grounded in epidemiological domain knowledge. This design contributes to both the interpretability and stability of its predictions. The effectiveness of the model was validated through experiments on real-world COVID-19 datasets, where it demonstrated competitive predictive performance compared to established benchmark models for multivariate time series forecasting. Although not always the top performer, PISID consistently ranked among the top two models across all forecasting scenarios—despite variations in regional scope, prevalent strains, and forecast horizons—demonstrating stable and reliable forecasting capabilities. We also conducted a comparative analysis of different encoder architectures and confirmed that information related to future epidemic dynamics can be effectively captured by modeling temporal dependencies using fully connected layers with residual connections, and spatial dependencies using a spatial embedding matrix. This architecture achieved an average improvement of 7.4% in MAE and 5.8% in RMSE compared to the best-performing baseline encoders, highlighting its effectiveness in epidemic forecasting. Furthermore, we demonstrated the interpretability of the model through a case study, highlighting how the explicit trajectory representation of the SIR module can provide meaningful insights into epidemic dynamics. In future work, we plan to incorporate external data related to epidemics—such as mobility patterns, distribution of viral strains, and vaccination rates—to more effectively capture shifts in epidemic dynamics in a timely manner. Regarding graph structure learning, this study raised concerns about performance degradation due to its complexity, but we believe that pursuing this direction remains valuable, given the spatial nature of infectious disease spread. To gain

a deeper understanding of the underlying trends in epidemic propagation, we aim to incorporate external data with richer spatio-temporal correlations and apply dynamic graph structure learning to explore transmission routes and delay patterns. We also plan to extend the SIR module to its variant forms by incorporating external data and introducing finer-grained compartments capable of disentangling and explaining individual contributing factors. This will enable a deeper integration of epidemiological knowledge into the neural network framework, ultimately supporting the development of public health and medical strategies.

## Acknowledgments

We thank all individuals who contributed indirectly to this research through discussions and insights.

## Author contributions

**Conceptualization:** Satoki Fujita.

**Funding acquisition:** Tatsuya Akutsu.

**Investigation:** Satoki Fujita.

**Methodology:** Satoki Fujita.

**Resources:** Tatsuya Akutsu.

**Software:** Satoki Fujita.

**Supervision:** Tatsuya Akutsu.

**Validation:** Satoki Fujita.

**Visualization:** Satoki Fujita.

**Writing – original draft:** Satoki Fujita.

**Writing – review & editing:** Tatsuya Akutsu.

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
