## [Decision Letter · Decision Letter 0]

9 Jul 2025

Dear Dr. Fujita,

Thank you for submitting your manuscript to PLOS ONE. After careful consideration, we feel that it has merit but does not fully meet PLOS ONE’s publication criteria as it currently stands. Therefore, we invite you to submit a revised version of the manuscript that addresses the points raised during the review process.

We look forward to receiving your revised manuscript.

Kind regards,

Guangyin Jin

Academic Editor

PLOS ONE

Journal Requirements: 

[SF is an employee at Shionogi & Co. There was no involvement of Shionogi & Co in the publication process. The other author declares no conflicts of interest.]. 

We note that one or more of the authors have an affiliation to the commercial funders of this research study : [Shionogi & Co].

Within your Competing Interests Statement, please confirm that this commercial affiliation does not alter your adherence to all PLOS ONE policies on sharing data and materials by including the following statement: ""This does not alter our adherence to  PLOS ONE policies on sharing data and materials.” (as detailed online in our guide for authors http://journals.plos.org/plosone/s/competing-interests). If this adherence statement is not accurate and  there are restrictions on sharing of data and/or materials, please state these. Please note that we cannot proceed with consideration of your article until this information has been declared.

Reviewers' comments:

Reviewer's Responses to Questions

**Comments to the Author**

1. Is the manuscript technically sound, and do the data support the conclusions?

Reviewer #1: Yes

Reviewer #2: Yes

2. Has the statistical analysis been performed appropriately and rigorously?

Reviewer #1: Yes

Reviewer #2: Yes

3. Have the authors made all data underlying the findings in their manuscript fully available?

Reviewer #1: Yes

Reviewer #2: Yes

4. Is the manuscript presented in an intelligible fashion and written in standard English?

Reviewer #1: Yes

Reviewer #2: Yes

Reviewer #1: 1- Add numeric values for model efficiency metrics to the abstract.

2- Using research papers published in 2025 and adding them to the related works.

3- Explaining the proposed algorithm in clear and simple steps for easy follow-up

4- Discuss tables and graphs better and reflect a full understanding of what is happening.

5- For conclusions not based on direct numerical values and future work not sufficiently explained

Reviewer #2: This study proposes a physics-informed spatial identity neural network for time-series forecasting of epidemic cases. The method integrates spatiotemporal information with domain knowledge for forecasting, and its performance is demonstrated using real data collected from Japan and the USA. The following comments are provided for the authors’ consideration:

1.The prediction performance heavily depends on the partitioning of datasets, and the proposed method is primarily applicable to time-series data with stationary patterns. This is evident from the results in Table 1 and Table 2. The proposed method does not consistently outperform others in all cases. The authors should highlight the limitations of the study and explain potential areas for improvement.

2.Lines 86–87: The key assumption for predicting time-series data is that there is no regime change in the dataset. Once the data becomes non-stationary, forward predictions become challenging. The authors should discuss how to detect regime or pattern changes.

3.Line 108: The integration of deep learning models (e.g., GNN) with epidemiological domain models has been explored by previous studies. The authors should clearly highlight the key methodological contributions of this work.

4.Line 146: How is the dimension of the latent space determined? The authors should also discuss the sensitivity of the results to this dimension.

5.Line 227: The data is pre-processed to remove seasonality. In practice, it is very difficult to detect periodicity in the early stages of prediction. The authors should elaborate on how this was handled.

6.The following references should be cited to strengthen the study’s context and relevance:

oTime series analysis and gated recurrent neural network model for predicting landslide displacements

oEnsemble learning of soil–water characteristic curve for unsaturated seepage using physics-informed neural networks

oInterpretation of spatio-temporal variation of precipitation from spatially sparse measurements using Bayesian compressive sensing (BCS)

**Do you want your identity to be public for this peer review?** For information about this choice, including consent withdrawal, please see our Privacy Policy

Reviewer #1: No

Reviewer #2: No

---

## [Author Response · Author response to Decision Letter 1]

10 Aug 2025

We sincerely thank the editor and the reviewrs for their thoughtful and constructive feedback on our manuscript. We have carefully considered all comments and revised the manuscript accordingly.

A detailed point-by-point response to each reviewer's comment is provided in the separate file titled “Response to Reviewers.docx”.

In response to the comments regarding the Funding Statement and Competing Interests Statement, we have revised them and included the updated versions in the cover letter.

---

## [Decision Letter · Decision Letter 1]

19 Aug 2025

Enhancing epidemic forecasting with a physics-informed spatial identity neural network

PONE-D-25-28281R1

Dear Dr. Fujita,

We’re pleased to inform you that your manuscript has been judged scientifically suitable for publication and will be formally accepted for publication once it meets all outstanding technical requirements.

Kind regards,

Guangyin Jin

Academic Editor

PLOS ONE

Additional Editor Comments (optional):

Reviewers' comments:

Reviewer's Responses to Questions

**Comments to the Author**

Reviewer #1: (No Response)

Reviewer #2: (No Response)

2. Is the manuscript technically sound, and do the data support the conclusions?

Reviewer #1: (No Response)

Reviewer #2: (No Response)

3. Has the statistical analysis been performed appropriately and rigorously?

Reviewer #1: (No Response)

Reviewer #2: (No Response)

4. Have the authors made all data underlying the findings in their manuscript fully available?

Reviewer #1: (No Response)

Reviewer #2: (No Response)

5. Is the manuscript presented in an intelligible fashion and written in standard English?

Reviewer #1: (No Response)

Reviewer #2: (No Response)

Reviewer #1: (No Response)

Reviewer #2: (No Response)

**Do you want your identity to be public for this peer review?** For information about this choice, including consent withdrawal, please see our Privacy Policy

Reviewer #1: No

Reviewer #2: No

---

## [Editor Report · Acceptance letter]

PONE-D-25-28281R1

PLOS ONE

Dear Dr. Fujita,

I'm pleased to inform you that your manuscript has been deemed suitable for publication in PLOS ONE. Congratulations! Your manuscript is now being handed over to our production team.

Kind regards,

on behalf of

Dr. Guangyin Jin

Academic Editor

PLOS ONE